# DYNAMIC LEAST-SQUARES REGRESSION

## ABSTRACT

In large-scale supervised learning, after a model is trained with an initial dataset, a common challenge is how to exploit new incremental data without re-training the model from scratch. Motivated by this problem, we revisit the canonical problem of *dynamic* least-squares regression (LSR), where the goal is to learn a linear model over incremental training data. In this setup, data and labels $(\mathbf{A}^{(t)}, \mathbf{b}^{(t)}) \in \mathbb{R}^{t \times d} \times \mathbb{R}^t$ evolve in an online fashion ($t \gg d$), and the goal is to efficiently *maintain* an (approximate) solution of $\min_{\mathbf{x}^{(t)}} \|\mathbf{A}^{(t)}\mathbf{x}^{(t)} - \mathbf{b}^{(t)}\|_2$ for all $t \in [T]$. Our main result is a dynamic data structure which maintains an arbitrarily small constant approximate solution to dynamic LSR with amortized update time $O(d^{1+o(1)})$, almost matching the running time of the *static* (sketching-based) solution. By contrast, for *exact* (or $1/\operatorname{poly}(n)$-accuracy) solutions, we show a separation between the models, namely, that dynamic LSR requires $\Omega(d^{2-o(1)})$ amortized update time under the OMv Conjecture (Henzinger et al., STOC'15). Our data structure is fast, conceptually simple, easy to implement, and our experiments demonstrate their practicality on both synthetic and real-world datasets.

## 1 INTRODUCTION

The problem of least-squares regression (LSR) dates back to Gauss in 1821 (Stigler, 1981), and is the backbone of statistical inference (Hastie et al., 2001), signal processing (Rabiner & Gold, 1975), convex optimization (Bubeck, 2015), control theory (Chui, 1990) and network routing (Lee & Sidford, 2014; Madry, 2013). Given an overdetermined ($n \gg d$) linear system $\mathbf{A} \in \mathbb{R}^{n \times d}, \mathbf{b} \in \mathbb{R}^n$, the goal is to find the solution vector $\mathbf{x}$ that minimizes the mean squared error (MSE)

$$\min_{\mathbf{x} \in \mathbb{R}^n} \|\mathbf{A}\mathbf{x} - \mathbf{b}\|_2. \tag{1}$$

Among many other loss functions (e.g., $\ell_p$) that have been studied for linear regression, $\ell_2$-regression has been the most popular choice as it is at the same time robust to outliers, and admits a *high-accuracy* efficient solution.

The computational task of least-squares regression arises naturally in high-dimensional statistics and has been the central of focus. The exact closed-form solution is given by the well-known Normal equation $\mathbf{x}^\star = (\mathbf{A}^\top \mathbf{A})^{-1} \mathbf{A}^\top \mathbf{b}$, which requires $O(nd^2)$ time to compute using naive matrix-multiplication, or $O(nd^{\omega-1}) \approx O(nd^{1.37})$ time using fast matrix-multiplication (FMM) (Strassen, 1969) for the current FMM exponent of $\omega \approx 2.37$ (Le Gall, 2014; Alman & Williams, 2021).

Despite the elegance and simplicity of this closed-form solution, in practice the latter runtime is often too slow, especially in modern data analysis applications where both the dimension of the feature space ($d$) and the size of datasets ($n$) are overwhelmingly large. A more modest objective in attempt to circumvent this computational overhead, is to seek an $\epsilon$-accurate solution that satisfies

$$\|\mathbf{A}\mathbf{x} - \mathbf{b}\|_2 \leq (1 + \epsilon) \min_{\mathbf{x} \in \mathbb{R}^d} \|\mathbf{A}\mathbf{x} - \mathbf{b}\|_2.$$

This was the primary motivation behind the development of the *sketch-and-solve* paradigm, where the idea is to first compress the matrix into one with fewer ($\sim d/\epsilon^2$) rows and then to compute the standard LSR solution but over the smaller matrix. A long line of developments on this framework culminates in algorithms that run in close to *input-sparsity* time (Sarlos, 2006; Clarkson & Woodruff, 2017; Nelson & Nguyên, 2013; Chepurko et al., 2021). In particular, a direct application of sketch-and-solve yields an algorithm runs in $\widetilde{O}(\operatorname{nnz}(A)\epsilon^{-1} + d^\omega)$ [1], which is near optimal in the "low

---

[1] In this paper we use $\widetilde{O}(\cdot)$ to hide polylogarithmic terms, and we use $O_\epsilon(\cdot)$ to hide $\operatorname{poly}(\log d, \epsilon^{-1})$ terms.

precision" regime ($\epsilon = 1/\operatorname{poly}(\log d)$). Interestingly, when combined with more sophisticated ideas of preconditioning and (conjugate) gradient descent, the runtime of this algorithm in terms of the error $\epsilon$ can be further improved to $\widetilde{O}(\operatorname{nnz}(A)\log(1/\epsilon) + d^\omega)$, which yields a *high precision* algorithm, i.e., it can efficiently solve the problem to within *polynomial accuracy* $\epsilon = 1/\operatorname{poly}(d)$.

**Dynamic least-squares**    In many real-world scenarios of the aforementioned applications, data is evolving in an online fashion either by nature or by design, and such applications require *maintaining* the solution (1) adaptively, where rows of the data matrix and their corresponding labels $(\mathbf{A}^{(t)}, \mathbf{b}^{(t)})$ arrive one-by-one *incrementally*. This is known as the dynamic least-squares regression problem.

The origins of dynamic least-squares regression was in control theory of the 1950's (Plackett, 1950), in the context of *dynamical linear systems*. In this setup, the data matrix $[\mathbf{A}^{(t)}, \mathbf{b}^{(t)}]$ corresponds to the set of measurement and it evolves in an online (incremental) fashion, and the goal is to efficiently maintain the (exact) solution to a noisy linear system $\mathbf{b} := \mathbf{A}^{(t)}\mathbf{x}^{(t)} + \xi^{(t)}$ *without* recomputing the LSR solution from scratch. The *recursive least-squares* (RLS) framework and the celebrated *Kalman filter* (Kalman, 1960) provide a rather simple update rule for maintaining an *exact* solution for this problem, by maintaining the sample covariance matrix and using *Woodburry's identity* (which assert that an incremental update to $[\mathbf{A}^{(t)}, \mathbf{b}^{(t)}]$ translates into a rank-1 update to the sample covariance matrix), and hence each update can be computed in $O(d^2)$ time (Kalman, 1960).

Beyond this classic motivation for dynamic LSR, a more timely motivation comes from modern deep learning applications: Most neural networks need to be frequently re-trained upon arrival on new training data, in order to improve prediction accuracy, and it is desirable to *avoid recomputing* weights from scratch. This problem of efficient incremental training of DNNs has been studied before in *elastic machine learning* (Liberty et al., 2020) and in the context of *continual learning* (Parisi et al., 2019). Our work sheds light on this question by analyzing the minimal computational resources required for $\ell_2$ loss-minimization.

Despite the rich and versatile literature on static LSR, the understanding of the dynamic counterpart was so far quite limited: The previous best known result requires $O(d^2)$ amortized update time (by a direct application of the Woodbury identity). The basic questions we address in this papers are:

*Is it possible to achieve faster update time for maintaining an exact solution? How about a small-approximate solution – Is it then possible to achieve amortized $O(d)$ or even input-sparsity time?*

In this paper, we settle both of these questions and present an essentially complete characterization of the dynamic complexity of LSR.

### 1.1    OVERVIEW OF OUR RESULTS

Our first result is a negative answer to the first question above of maintaining exact (or polynomial-accuracy) LSR solutions in the dynamic setting – We prove that Kalman's approach is essentially optimal, assuming the popular *Online Matrix-Vector* (OMv) Conjecture (Henzinger et al., 2015) [2]:

**Theorem 1.1** (Hardness of exact dynamic LSR, informal). *Assuming the OMv Conjecture, any dynamic algorithm that maintains an $\epsilon = 1/\operatorname{poly}(d)$-approximate solution for the dynamic LSR problem over $T = \operatorname{poly}(d)$ iterations, must have $\Omega(d^{2-o(1)})$ amortized update time per iteration.*

Theorem 1.1 separates the static and the dynamic complexities of the exact LSR problem: As mentioned above, the static problem can be solved by *batching* rows together using FMM in time $O(Td^{\omega-1})$, whereas the dynamic problem requires $\Omega(Td^2)$ by Theorem 1.1. Indeed, the implication Theorem 1.1 is stronger, it also separates the static and dynamic complexity of approximate LSR problem under the high precision regime, it asserts that a polylogarithmic dependence on the precision (i.e. $d\operatorname{poly}(\log(1/\epsilon))$) on update time is impossible (assuming OMv), in sharp contrast to the static case.

We next focus on an *approximate* version of this classic online problem, *dynamic $\epsilon$-LSR*, where the goal is to efficiently maintain, during *all* iterations $t \in [T]$, an $\epsilon$-approximate solution under incremental row-updates to $\mathbf{A}^{(t)}$ and labels $\mathbf{b}^{(t)}$, where efficiency is measured by the *amortized*

---

[2]This conjecture postulates that multiplying a fixed $d \times d$ matrix $A$ with an *online* matrix $B$, column-by-column ($AB_i$), requires $d^{3-o(1)}$ time, in sharp contrast to the batch setting where this can be done using FMM in $d^\omega \ll d^3$ time. See Section 4.

*update time* for inserting a new row. A natural complexity benchmark for this dynamic problem is the aforementioned best *static* sketch-and-solve solution, which for $n = T$ is $\widetilde{O}(\mathrm{nnz}(\mathbf{A}^{(T)})\epsilon^{-1} + d^\omega) = \widetilde{O}(\mathrm{nnz}(\mathbf{A})\epsilon^{-1})$ for $T \gg d$. Our main result is a provably efficient and practical dynamic data structure, whose total running time essentially matches the complexity of the offline problem:

**Theorem 1.2** (Main result, informal version of Theorem 3.1). *For any accuracy parameter $\epsilon > 0$, there is a randomized dynamic data structure which, with probability at least $0.9$, maintains an $\epsilon$-approximate solution to the dynamic LSR problem simultaneously for all iterations $t \in [T]$, with total update time*

$$O(\epsilon^{-2} \mathrm{nnz}(\mathbf{A}^{(T)}) \log(T) + \epsilon^{-6} d^3 \log^5(T)).$$

Theorem 1.2 almost matches the fastest static sketching-based solution, up to polylogarithmic terms and the additive FMM term. When $T \gg d$, this theorem shows that amortized update time of our algorithm is $O(d^{1+o(1)})$.

## 1.2 RELATED WORK

**Sketching and sampling** The least squares regression as a fundamental problem has been extensively studied in the literature. A long line of work (Ailon & Chazelle, 2006; Clarkson & Woodruff, 2017; Nelson & Nguyên, 2013; Avron et al., 2017; Cohen et al., 2015; Woodruff, 2014; 2021) have focused on using dimension reduction technique (sketching or sampling) to speedup the computation task, culminates into algorithms that run in $\widetilde{O}(\mathrm{nnz}(\mathbf{A}) \log(1/\epsilon) + d^\omega)$ time (Clarkson & Woodruff, 2017). See Appendix A for detailed discussions.

**Regression in online, streaming, and sliding window models** Least-squares regressions have also been studied in various computational models, though the focus of these models are generally not the (amortized) running time. Our algorithm uses techniques developed by Cohen et al. (2020), where they study the regression problem in the online model, with the goal of maintaining a spectral approximation of data matrix in the online stream. Their algorithm only needs to store $O_\epsilon(d)$ rows but the amortized running time is still $\Omega(d^2)$ (see Section 3.1 for detailed discussions). In the streaming model, the main focus is the space complexity, and a direct application of random Gaussian sketch or count sketch reduces the space complexity to $\widetilde{O}(d^2\epsilon^{-1})$ and it is shown to be tight (Clarkson & Woodruff, 2009). Recent work of (Braverman et al., 2020) studies regressions and other numerical linear algebra tasks in the sliding window model, where data come in an online stream and only the most recent updates form the underlying data set. The major focus of a sliding window model is still the space complexity, and there is no amortized running time guarantee.

**Disparity from online learning** Our work crucially differs from online learning literature (Hazan, 2019), in that the main bottleneck in online regret-minimization and bandit problems is information-theoretic, whereas the challenge in our loss-minimization problem is purely computational. See Appendix A for detailed discussions.

## 2 PROBLEM FORMULATION

In a dynamic least-squares regression problem, initially, we are given a matrix $\mathbf{A}^{(0)} \in \mathbb{R}^{n_0 \times d}$ together with a vector $\mathbf{b}^{(0)} \in \mathbb{R}^{n_0}$. At the $t$-th step, a new data of form $((\mathbf{a}^{(t)})^\top, \beta^{(t)}) \in \mathbb{R}^d \times \mathbb{R}$ arrives, and the goal is to maintain an $\epsilon$-approximate solution. A formal description is provided below, where we assume $n_0 = d + 1$ for simplicity (see Remark 2.3).

**Definition 2.1** (Dynamic least-squares regression). *Let $d \in \mathbb{N}_+$ and $\epsilon \in [0, 1)$ be two fixed parameters. We say an algorithm solves $\epsilon$-approximate dynamic least squares regression if*

- *The data structure is given a matrix $\mathbf{A}^{(0)} \in \mathbb{R}^{(d+1) \times d}$ and a vector $\mathbf{b}^{(0)} \in \mathbb{R}^{d+1}$ in the preprocessing phase.*

- *For each iteration $t \in [T]$, the algorithm receives updates $\mathbf{a}^{(t)} \in \mathbb{R}^d$ and $\beta^{(t)} \in \mathbb{R}$. Define $\mathbf{A}^{(t)} := [(\mathbf{A}^{(t-1)})^\top, \mathbf{a}^{(t)}]^\top \in \mathbb{R}^{(d+t+1) \times d}$ to be $\mathbf{A}^{(t-1)}$ appended with a new row $(\mathbf{a}^{(t)})^\top$, and $\mathbf{b}^{(t)} := [(\mathbf{b}^{(t-1)})^\top, \beta^{(t)}]^\top \in \mathbb{R}^{d+t+1}$ to be $\mathbf{b}^{(t-1)}$ appended with a new entry $\beta^{(t)}$. After this update, the algorithm outputs an $\epsilon$-approximate solution $\mathbf{x}^{(t)} \in \mathbb{R}^d$:*

$$\|\mathbf{A}^{(t)}\mathbf{x}^{(t)} - \mathbf{b}^{(t)}\|_2 \leq (1 + \epsilon) \min_{\mathbf{x} \in \mathbb{R}^d} \|\mathbf{A}^{(t)}\mathbf{x} - \mathbf{b}^{(t)}\|_2.$$

We write $[0:T] = \{0, 1, \ldots, T\}$, and for any $t \in [0:T]$, we denote $\mathbf{M}^{(t)} := [\mathbf{A}^{(t)}, \mathbf{b}^{(t)}] \in \mathbb{R}^{(d+t+1)\times(d+1)}$. We make the following assumptions.

**Assumption 2.2.** *We assume 1. Each data have bounded $\ell_2$ norm, i.e., $\forall i \in [T+d+1]$, the $i$-th row of $\mathbf{M}^{(T)}$ satisfies $\|\mathbf{M}_{i,*}^{(T)}\|_2 \leq D$. 2. The initial matrix $\mathbf{M}^{(0)}$ has full rank, and its smallest singular value is bounded by $\sigma_{d+1}(\mathbf{M}^{(0)}) \geq \sigma$ for some polynomially small $\sigma \in (0,1)$.*

**Remark 2.3.** *We remark that these assumptions are essentially w.l.o.g. for the following reasons: 1. Real world data inherently have bounded $\ell_2$ norm, and in applications like machine learning, data are often normalized. 2. We can assume the initial matrix $\mathbf{A}^{(0)}$ has $d+1$ rows because brute-forcely adding these $d+1$ initial rows would only take $O(d^3)$ time, and this is within our desired total running time of $O_\epsilon(\mathrm{nnz}(\mathbf{A}^{(T)}) + d^3)$. 3. To satisfy the assumption that $\sigma_{d+1}(\mathbf{M}^{(0)}) \geq \sigma$ for some polynomially small $\sigma$, we could let the initial matrix $\mathbf{M}^{(0)} = \sigma \cdot \mathbf{I}_{d+1}$. This is equivalent to adding a small regularization term of $\sigma \cdot \|\mathbf{x}\|_2$ and this incurs only a polynomially small additive error.*

## 3 DYNAMIC $\epsilon$-LSR DATA STRUCTURE

In this section, we provide an approximation algorithm for the dynamic least squares regression. Notably, our algorithm maintains an $\epsilon$-approximate solution in near input sparsity time.

**Theorem 3.1** (Data structure for dynamic least squares regression). *Let $\epsilon > 0$, $d, T \in \mathbb{N}$. There exists a randomized algorithm for dynamic least-squares regression (Algorithm 1–4). With probability at least $0.9$, the algorithm maintains an $\epsilon$-approximation solution for all iterations $t \in [T]$ and the total update time over $T$ iterations is at most $O(\epsilon^{-2} \mathrm{nnz}(\mathbf{A}^{(T)}) \log(T) + \epsilon^{-6} d^3 \log^5(TD/\sigma))$. Our data structure uses at most $O(\epsilon^{-2} d^2 \cdot \log^2(TD/\sigma))$ space.*

**Notations** We use a superscript $^{(t)}$ to denote the matrix/vector/scaler maintained by the data structure at the end of the $t$-th iterations. In particular, the superscript $^{(0)}$ represents the variables after the preprocessing step. For any matrix $\mathbf{A} \in \mathbb{R}^{n\times d}$, $i \in [n]$ we define its leverage score $\tau(\mathbf{A}) \in \mathbb{R}^n$ as $\tau_i(\mathbf{A}) := \mathbf{a}_i^\top (\mathbf{A}^\top \mathbf{A})^\dagger \mathbf{a}_i$. We define the generalized leverage score (same as Cohen et al. (2015)) of $\mathbf{A} \in \mathbb{R}^{n\times d}$ with respect to another matrix $\mathbf{B} \in \mathbb{R}^{n'\times d}$ as: $\tau_i^{\mathbf{B}}(\mathbf{A}) := \mathbf{a}_i^\top (\mathbf{B}^\top \mathbf{B})^\dagger \mathbf{a}_i$. For more properties of the leverage scores see Section B.1.

### 3.1 TECHNIQUE OVERVIEW

Our approach is formally described in Algorithm 1–4, we first overview the ideas behind it.

From a high level view, our approach follows the online row sampling framework (Cohen et al., 2015; 2020; Braverman et al., 2020): When a new row arrives, we sample and keep the new row with probability proportional to the (approximated version of) *online leverage score*

$$\tau_{d+t+1}^{\mathbf{M}^{(t-1)}}(\mathbf{M}^{(t)}) = (\mathbf{m}^{(t)})^\top ((\mathbf{M}^{(t-1)})^\top \mathbf{M}^{(t-1)})^{-1} \mathbf{m}^{(t)}.$$

The sampled matrix is a spectral approximation to the true data matrix. We maintain an approximate least-squares regression solution using this sampled matrix.

Naively computing the online leverage score takes $O(d^2)$ time. In order to accelerate this computation, we use two approximations:

1. Similar to Cohen et al. (2020), we compute the online leverage scores with respect to the sampled matrix instead of the true data matrix. However, this idea alone is still not enough to achieve sub-quadratic time.

2. We use a JL-embedding[3] trick (Spielman & Srivastava, 2011) to compress the size of the $d \times d$ matrix to $\approx \epsilon^{-2} \times d$. In this way, in each iteration it only takes $O_\epsilon(d)$ time to compute the approximate online leverage score.

We further use an inductive analysis to bound the overall error (Lemma 3.4).

Finally, we adopt a similar strategy as Cohen et al. (2020) to prove that sampling according to the approximate online leverage score still keeps at most $O_\epsilon(d)$ sampled rows (Lemma 3.7). Whenever a row is sampled, it takes $O(d^2)$ time to update the maintained matrices using Woodbury identity. Hence, the amortized update time of the sampled rows is $O_\epsilon(d^3/T) = o(d)$ when $T \gg d$.

---

[3]Johnson-Lindenstrauss (JL) Lemma shows a way to embed high-dimensional vectors to low-dimensional space while preserving the distances between the vectors. A rigorous statement is shown in Lemma B.4.

**Remark 3.2** (Difference from sketching-based solutions). *Our approach crucially differs from the sketching-based solutions, which do not provide any speedup over the direct application of Woodburry identity ($O(d^2)$ time per iteration). A sketching-based solution maintains a sketched matrix $\mathbf{SM} \in \mathbb{R}^{O_\epsilon(d) \times d}$, where $\mathbf{S}$ is a sketching matrix (e.g. SRHT (Ailon & Chazelle, 2006) or Count Sketch (Clarkson & Woodruff, 2017)) that mixes the rows of $\mathbf{M}$. When a new row of $\mathbf{M}$ arrives, at least one row of the sketched matrix $\mathbf{SM}$ needs to be updated, in contrast to our sampling-based approach where the sampled matrix is not updated in most of the iterations.*

**Implementation** We explain the detailed implementation of our algorithm. At the beginning of the $t$-th iteration, a sampling matrix $\mathbf{D}^{(t-1)}$ is derived based on the online leverage score, and the sub-sampled matrix $\mathbf{N}^{(t-1)} = \mathbf{D}^{(t-1)}\mathbf{M}^{(t-1)}$ maintains a spectral approximation on the column space of $\mathbf{M}^{(t-1)} = [\mathbf{A}^{(t-1)}, \mathbf{b}^{(t-1)}]$. Let $s^{(t-1)}$ denote the number of sampled rows. To obtain spectral approximation, we maintain the approximate covariance matrices $\mathbf{H}^{(t-1)} = ((\mathbf{N}^{(t-1)})^\top \mathbf{N}^{(t-1)})^{-1}$ and $\mathbf{B}^{(t-1)} = \mathbf{N}^{(t-1)}\mathbf{H}^{(t-1)}$. The online leverage score $\tau^{(t)}$ of a new row $\mathbf{m}^{(t)} = ((\mathbf{a}^{(t)})^\top, \beta^{(t)})$ can be approximated as $\|\mathbf{B}^{(t-1)}\mathbf{m}^{(t-1)}\|_2$. To efficiently compute the leverage score of a new row, we left-multiply by a JL matrix $\mathbf{J}^{(t-1)}$ and maintain a proxy $\widetilde{\mathbf{B}}^{(t-1)} = \mathbf{J}^{(t-1)}\mathbf{B}^{(t-1)}$, with the guarantee that $\|\widetilde{\mathbf{B}}^{(t-1)}\mathbf{m}\|_2 \approx \|\mathbf{B}^{(t-1)}\mathbf{m}\|_2$ for any $\mathbf{m} \in \mathbb{R}^{d+1}$ with high probability.

When a new row $\mathbf{m}^{(t)}$ is inserted at the $t$-th iteration, we sample it via the approximate online leverage score (Line 3 of SAMPLE). We only perform update if the new row is sampled. In that case, we renew the JL matrix and perform a series of careful updates on all the variables that we maintain (See UPDATEMEMBERS). To obtain the final solution $\mathbf{x}^{(t)} \in \mathbb{R}^d$, we solve $\min_{\mathbf{x} \in \mathbb{R}^d} \|\mathbf{D}^{(t)}\mathbf{A}^{(t)}\mathbf{x} - \mathbf{D}^{(t)}\mathbf{b}^{(t)}\|_2$, which has the closed-form solution of $\mathbf{x}^{(t)} = \mathbf{G}^{(t)} \cdot \mathbf{u}^{(t)}$ and can be efficiently maintained by taking $\mathbf{G}^{(t)} = ((\mathbf{A}^{(t)})^\top (\mathbf{D}^{(t)})^2 \mathbf{A}^{(t)})^{-1} \in \mathbb{R}^{d \times d}$ and $\mathbf{u}^{(t)} = (\mathbf{A}^{(t)})^\top (\mathbf{D}^{(t)})^2 \mathbf{b}^{(t)}$.

---

**Algorithm 1** PREPROCESS $(\mathbf{A}, \mathbf{b}, \epsilon, T)$

1: $\mathbf{M} \leftarrow [\mathbf{A}, \mathbf{b}]$      ▷ $\mathbf{M} \in \mathbb{R}^{(d+1) \times (d+1)}$
2: $\mathbf{D} \leftarrow \mathbf{I}_{d+1}$
   # Spectral approximation
3: $s \leftarrow d + 1$
4: $\mathbf{N} \leftarrow \mathbf{D} \cdot \mathbf{M}$      ▷ $\mathbf{N} \in \mathbb{R}^{s \times (d+1)}$
5: $\mathbf{H} \leftarrow ((\mathbf{N})^\top \mathbf{N})^{-1}$   ▷ $\mathbf{H} \in \mathbb{R}^{(d+1) \times (d+1)}$
6: $\mathbf{B} \leftarrow \mathbf{N} \cdot \mathbf{H}$      ▷ $\mathbf{B} \in \mathbb{R}^{s \times (d+1)}$
   # JL approximation
7: $\delta \leftarrow O(1/T^2)$, $k \leftarrow O(\epsilon^{-2}\log(T/\delta))$
8: $\mathbf{J} \leftarrow \text{JL}(s, \epsilon, \delta, T)$ ▷ JL matrix $\mathbf{J} \in \mathbb{R}^{k \times s}$
9: $\widetilde{\mathbf{B}} \leftarrow \mathbf{J} \cdot \mathbf{B}$      ▷ $\widetilde{\mathbf{B}} \in \mathbb{R}^{k \times (d+1)}$
   # Maintain solution
10: $\mathbf{G} \leftarrow (\mathbf{A}^\top \mathbf{D}^\top \mathbf{D}\mathbf{A})^{-1}$    ▷ $\mathbf{G} \in \mathbb{R}^{d \times d}$
11: $\mathbf{u} \leftarrow \mathbf{A}^\top \mathbf{D}^2 \mathbf{b}$      ▷ $\mathbf{u} \in \mathbb{R}^d$
12: $\mathbf{x} \leftarrow \mathbf{G} \cdot \mathbf{u}$      ▷ $\mathbf{x} \in \mathbb{R}^d$

---

**Algorithm 2** UPDATE $(\mathbf{a}, \beta)$

1: $\mathbf{m} \leftarrow [\mathbf{a}^\top, \beta]^\top$      ▷ $\mathbf{m} \in \mathbb{R}^{d+1}$
2: $\nu \leftarrow$ SAMPLE$(\mathbf{m})$      ▷ $\nu \in \mathbb{R}$
3: $\mathbf{D} \leftarrow \begin{bmatrix} \mathbf{D} & 0 \\ 0 & \nu \end{bmatrix}$
4: **if** $\nu \neq 0$ **then** UPDATEMEMBERS$(\mathbf{m})$
5: **return** $\mathbf{x}$

---

**Algorithm 3** SAMPLE $(\mathbf{m})$

1: $\tau \leftarrow \|\widetilde{\mathbf{B}} \cdot \mathbf{m}\|_2^2$
2: $p \leftarrow \min\{3(1 + \epsilon)^2 \epsilon^{-2} \tau \log(1/\delta), 1\}$
3: $\nu \leftarrow 1/\sqrt{p}$ with probability $p$, and $\nu \leftarrow 0$ otherwise

---

**Algorithm 4** UPDATEMEMBERS $(\mathbf{m})$

   # Update spectral approximation
1: $s \leftarrow s + 1$
2: $\Delta\mathbf{H} \leftarrow -\frac{\mathbf{H}\mathbf{m}\mathbf{m}^\top \mathbf{H}/p}{1 + \mathbf{m}^\top \mathbf{H}\mathbf{m}/p}$
3: $\mathbf{H} \leftarrow \mathbf{H} + \Delta\mathbf{H}$
4: $\mathbf{B} \leftarrow [(\mathbf{B} + \mathbf{N} \cdot \Delta\mathbf{H})^\top, \ \mathbf{H} \cdot \mathbf{m}/\sqrt{p}]^\top$
5: $\mathbf{N} \leftarrow [\mathbf{N}^\top, \mathbf{m}/\sqrt{p}]^\top$
   # Update JL approximation
6: $\mathbf{J} \leftarrow \text{JL}(s, \epsilon, \delta, T)$
7: $\widetilde{\mathbf{B}} \leftarrow \mathbf{J} \cdot \mathbf{B}$
   # Update solution
8: $\mathbf{G} \leftarrow \mathbf{G} - \frac{\mathbf{G}\mathbf{a}\mathbf{a}^\top \mathbf{G}/p}{1 + \mathbf{a}^\top \mathbf{G}\mathbf{a}/p}$
9: $\mathbf{u} \leftarrow \mathbf{u} + \beta \cdot \mathbf{a}/p$
10: $\mathbf{x} \leftarrow \mathbf{G} \cdot \mathbf{u}$

---

We outline the proof of Theorem 3.1, and defer the detailed proof to Appendix C due to space limits.

### 3.2 CORRECTNESS

We show the correctness of our algorithm and prove it maintains an $\epsilon$-approximate solution for all iterations with high probability. We start with closed-form formulas for all the variables we maintain.

**Lemma 3.3** (Closed-form formulas). *At the $t$-th iteration of Algorithm 1 – 4, we have*

1. $\mathbf{M}^{(t)} = [\mathbf{A}^{(t)}, \mathbf{b}^{(t)}] \in \mathbb{R}^{(d+t+1)\times(d+1)}$.

2. $\mathbf{D}^{(t)} \in \mathbb{R}^{(d+t+1)\times(d+t+1)}$ *is a diagonal matrix with $s^{(t)}$ non-zero entries.*

3. $\mathbf{N}^{(t)} = (\mathbf{D}^{(t)}\mathbf{M}^{(t)})_{S^{(t)},*} \in \mathbb{R}^{s^{(t)}\times(d+1)}$, *where $S^{(t)} \subset [d+t+1]$ is defined as the set of non-zero entries of $\mathbf{D}^{(t)}$.*

4. $\mathbf{H}^{(t)} = \big((\mathbf{N}^{(t)})^\top\mathbf{N}^{(t)}\big)^{-1} \in \mathbb{R}^{(d+1)\times(d+1)}$.

5. $\mathbf{B}^{(t)} = \mathbf{N}^{(t)}\mathbf{H}^{(t)} \in \mathbb{R}^{s^{(t)}\times(d+1)}$.

6. $\widetilde{\mathbf{B}}^{(t)} = \mathbf{J}^{(t)} \cdot \mathbf{B}^{(t)} \in \mathbb{R}^{k\times(d+1)}$, *where $k = O(\epsilon^{-2}\log(T/\delta))$.*

7. $\mathbf{G}^{(t)} = \big((\mathbf{A}^{(t)})^\top(\mathbf{D}^{(t)})^2\mathbf{A}^{(t)}\big)^{-1} \in \mathbb{R}^{d\times d}$.

8. $\mathbf{u}^{(t)} = (\mathbf{A}^{(t)})^\top(\mathbf{D}^{(t)})^2\mathbf{b}^{(t)} \in \mathbb{R}^d$.

9. $\mathbf{x}^{(t)} = \big((\mathbf{A}^{(t)})^\top(\mathbf{D}^{(t)})^2\mathbf{A}^{(t)}\big)^{-1} \cdot (\mathbf{A}^{(t)})^\top(\mathbf{D}^{(t)})^2\mathbf{b}^{(t)} \in \mathbb{R}^d$.

The following lemma is key for our correctness analysis. It shows that we maintain a good approximation on online leverage scores and a spectral approximation of $\mathbf{M}^{(t)}$ throughout all iterations.

**Lemma 3.4** (Spectral approximation via leverage score maintenance). *With probability at least $1 - 2T\delta$,*

$$(1-\epsilon)^2\tau_{d+t+1}^{\mathbf{M}^{(t-1)}}(\mathbf{M}^{(t)}) \leq \tau^{(t)} \leq (1+\epsilon)^2\tau_{d+t+1}^{\mathbf{M}^{(t-1)}}(\mathbf{M}^{(t)}), \ \ \forall t \in [T], \tag{2}$$

*and*

$$(\mathbf{M}^{(t)})^\top(\mathbf{D}^{(t)})^2\mathbf{M}^{(t)} \approx_\epsilon (\mathbf{M}^{(t)})^\top\mathbf{M}^{(t)}, \ \ \forall t \in [0:T]. \tag{3}$$

*Proof Sketch.* We prove by induction and show that with probability $1 - 2t\delta$, Eq. (3) holds for all $t' \in [0:t]$ and Eq. (2) holds for all $t' \in [t]$. The base case $t = 0$ holds trivially, as $\mathbf{D}^{(0)} = \mathbf{I}_{d+1}$, and therefore, $(\mathbf{M}^{(0)})^\top(\mathbf{D}^{(0)})^2\mathbf{M}^{(0)} = (\mathbf{M}^{(0)})^\top\mathbf{M}^{(t)}$. Given the induction hypothesis upon $t - 1$, we proceed in the following three steps.

- We first use the induction hypothesis to prove that $\|\mathbf{B}^{(t-1)}\mathbf{m}^{(t)}\|$ is a good estimate on the online leverage score, that is

$$(1-\epsilon)\tau_{d+t+1}^{\mathbf{M}^{(t-1)}}(\mathbf{M}^{(t)}) \leq \|\mathbf{B}^{(t-1)}\cdot\mathbf{m}^{(t)}\|_2^2 \leq (1+\epsilon)\tau_{d+t+1}^{\mathbf{M}^{(t-1)}}(\mathbf{M}^{(t)}).$$

- We then use the JL lemma (Lemma B.4) to show that with probability $1 - \delta$, the sketched covariance matrix $\|\widetilde{\mathbf{B}}^{(t-1)}\mathbf{m}^{(t)}\|$ returns good estimation $\|\mathbf{B}^{(t-1)}\mathbf{m}^{(t)}\|$. That is

$$(1-\epsilon)^2 \cdot \tau_{d+t+1}^{\mathbf{M}^{(t-1)}}(\mathbf{M}^{(t)}) \leq \|\widetilde{\mathbf{B}}^{(t-1)}\cdot\mathbf{m}^{(t)}\|_2^2 \leq (1+\epsilon)^2 \cdot \tau_{d+t+1}^{\mathbf{M}^{(t-1)}}(\mathbf{M}^{(t)}). \tag{4}$$

- Finally, we wrap up the proof by proving the second part of induction. In particular, we show that conditioned on Eq. (4) holds, we have

$$(\mathbf{M}^{(t)})^\top(\mathbf{D}^{(t)})^2\mathbf{M}^{(t)} \approx_\epsilon (\mathbf{M}^{(t)})^\top\mathbf{M}^{(t)}.$$

The proof then follows by an union bound over failure events. □

It is well known that spectral approximations of $(\mathbf{M}^{(t)})^\top\mathbf{M}^{(t)}$ give approximate solutions to least squares regressions (Woodruff, 2014), so we have proved the correctness of our algorithm.

**Lemma 3.5** (Correctness of Algorithm 1–4). *With probability at least $1 - O(1/T)$, in each iteration,* UPDATE *of Algorithm 2 outputs a vector $\mathbf{x}^{(t)} \in \mathbb{R}^d$ such that*

$$\|\mathbf{A}^{(t)}\mathbf{x}^{(t)} - \mathbf{b}^{(t)}\|_2 \leq (1+\epsilon)\min_{\mathbf{x}\in\mathbb{R}^d}\|\mathbf{A}^{(t)}\mathbf{x} - \mathbf{b}^{(t)}\|_2.$$

### 3.3 TIME ANALYSIS

Next, we bound the overall update time of our algorithm. We first compute the worst case update time of Algorithm 2. When $\nu^{(t)} = 0$, i.e., the $t$-th row is not sampled, the UPDATE procedure only needs to compute the approximate leverage score $\tau^{(t)}$ (SAMPLE, Algorithm 3), and it takes $O(k \cdot \mathrm{nnz}(\mathbf{m}^{(t)}))$ time. When $\nu^{(t)} \neq 0$, i.e., the $t$-th row is sampled, the UPDATE procedure makes a call to UPDATEMEMBERS (Algorithm 4), and it takes $O(k \cdot s^{(t)} \cdot d)$ time. Plugging in the value of $k$, we have the following lemma.

**Lemma 3.6** (Worst case update time). *At the $t$-th iteration of the* UPDATE *procedure (Algorithm 2),*

- *If $\nu^{(t)} = 0$, then* UPDATE *takes $O(\epsilon^{-2} \log(T/\delta) \cdot \mathrm{nnz}(\mathbf{a}^{(t)}))$ time.*

- *If $\nu^{(t)} \neq 0$, then* UPDATE *takes $O(\epsilon^{-2} s^{(t)} d \log(T/\delta))$ time.*

To bound the amortized update time, we need to bound the total number of sampled rows, and this is closely related to the sum of online leverage scores. Such an upper bound was already established by Cohen et al. (2020), here we present a slightly generalized version of it.

**Lemma 3.7** (Sum of online leverage scores, generalization of Theorem 2.2 of (Cohen et al., 2020)). *If the matrix $\mathbf{M}^{(T)}$ satisfy Assumption 2.2, then*

$$\sum_{t=1}^{T} \tau_{d+t+1}^{\mathbf{M}^{(t-1)}}(\mathbf{M}^{(t)}) \leq O(d \log(TD/\sigma)).$$

Now we are ready to bound the amortized update time of our algorithm.

**Lemma 3.8** (Amortized update time). *With probability at least $0.99$, the total running time of* UPDATE *over $T$ iterations is at most $O(\epsilon^{-2} \mathrm{nnz}(\mathbf{A}^{(T)}) \log(T) + \epsilon^{-6} d^3 \log^5(TD/\sigma))$.*

*Proof Sketch.* In this proof sketch we simplify the second term as $d^3 \cdot \mathrm{poly}(\epsilon^{-1} \log(TD/\sigma))$. The first term comes from the computation cost of querying leverage score, which takes $O(\epsilon^{-2} \log(T/\delta) \cdot \mathrm{nnz}(\mathbf{a}^{(t)}))$ time in the $t$-th iteration even if the $t$-th row is not sampled. The second term bounds the total update time for the sampled rows:

- From Lemma 3.4 and Lemma 3.7, with high probability the sum of the approximate online leverage scores $\tau^{(t)}$ are bounded by $O(d \log(TD/\sigma))$.

- Using Markov inequality, the total number of sampled rows is bounded by

$$s^{(T)} = O(\sum_{i=1}^{T} p^{(t)}) = O(\epsilon^{-2} \cdot \log(1/\delta) \cdot \sum_{t=1}^{T} \tau^{(t)}) \leq O(d \cdot \mathrm{poly}(\epsilon^{-1} \log(TD/\sigma))).$$

- Since there are $s^{(T)}$ sampled rows, and for each sampled row we update data structure members in $\leq O(s^{(T)} d \cdot \mathrm{poly}(\epsilon^{-1} \log(TD/\sigma)))$ time, the total update time for sampled rows is

$$s^{(T)} \cdot O(s^{(T)} d \cdot \mathrm{poly}(\epsilon^{-1} \log(TD/\sigma))) = O(d^3 \cdot \mathrm{poly}(\epsilon^{-1} \log(TD/\sigma))). \qquad \square$$

## 4 HARDNESS RESULT

We prove a $\Omega(d^{2-o(1)})$ amortized time lower bound for dynamic least squares regression with high precision, assuming the OMv conjecture. The OMv conjecture was originally proposed by Henzinger et al. (2015), and it is widely accepted in the theoretical computer science community.

**Conjecture 4.1** (OMv conjecture, (Henzinger et al., 2015)). *Let $d \in \mathbb{N}, T = \mathrm{poly}(d)$. Let $\gamma > 0$ be any constant. $\mathbf{B} \in \{0,1\}^{d \times d}$ is a Boolean matrix. $\forall t \in [T]$, a Boolean vector $\mathbf{z}^{(t)} \in \{0,1\}^d$ is revealed at the $t$-th step. We say an algorithm solves the OMv problem if it returns the Boolean matrix-vector product $\mathbf{B}\mathbf{z}^{(t)} \in \mathbb{R}^d$ at every time step. The conjectures states that there is no algorithm that solves the OMv problem using $\mathrm{poly}(d)$ preprocessing time and $O(d^{2-\gamma})$ amortized running time, and has an error probability $\leq 1/3$.*

The results in this section are all under the Word RAM model where the word size $w = O(\log d)$. Our main result is formally stated below.

**Theorem 4.2** (Hardness of dynamic-least squares regression with high precision). *Let $d \in \mathbb{N}$, $T = \text{poly}(d)$, $\epsilon = \frac{1}{d^8 T^2} = 1/\text{poly}(d)$, and let $\gamma > 0$ be any constant. Assuming the OMv conjecture is true, any dynamic algorithm that maintains an $\epsilon$-approximate solution of the least squares regression requires at least $\Omega(d^{2-\gamma})$ amortized time per update.*

Our lower bound is proved by first reducing the standard OMv conjecture for Boolean matrices to OMv-hardness for well-conditioned positive semidefinite (PSD) matrices over real numbers. Then we use this new OMv-hardness result to prove our lower bound for dynamic least squares regression. We only provide a proof sketch here and detailed proof are delayed to Section D.

**OMv-hardness for well-conditioned PSD matrix**    The OMv conjecture asserts the hardness of solving online Boolean matrix-vector product *exactly*. We extend it to solving online real-valued matrix-vector product for well-conditioned PSD matrices, while allowing polynomially small error.

**Lemma 4.3** (Hardness of approximate real-valued OMv). *Let $d \in \mathbb{N}, T = \text{poly}(d)$. Let $\gamma > 0$ be any constant. Let $\mathbf{H} \in \mathbb{R}^{d \times d}$ be a symmetric matrix whose eigenvalues satisfy $1 \leq \lambda_d(\mathbf{H}) \leq \cdots \leq \lambda_1(\mathbf{H}) \leq 3$. For any $t \in [T]$, $\mathbf{z}^{(t)} \in \mathbb{R}^d$ is revealed at the $t$-th step, and $\|\mathbf{z}^{(t)}\|_2 \leq 1$. Assuming the OMv conjecture is true, then there is no algorithm with $\text{poly}(d)$ preprocessing time and $O(d^{2-\gamma})$ amortized running time that can return an $O(1/d^2)$-approximate answer to $\mathbf{H}\mathbf{z}^{(t)}$ for all $t$, i.e., a vector $\mathbf{y}^{(t)} \in \mathbb{R}^d$ s.t. $\|\mathbf{y}^{(t)} - \mathbf{H}\mathbf{z}^{(t)}\|_2 \leq \epsilon$, and has an error probability $\leq 1/3$.*

*Proof Sketch.* Given a Boolean matrix $\mathbf{B} \in \{0, 1\}^{d \times d}$ in the OMv conjecture, we construct a PSD matrix $\mathbf{H} = \begin{bmatrix} 2\mathbf{I}_d & \frac{1}{d}\mathbf{B} \\ \frac{1}{d}\mathbf{B}^\top & 2\mathbf{I}_d \end{bmatrix} \in \mathbb{R}^{2d \times 2d}$. We note that $\mathbf{H}$ is symmetric and $1 \leq \lambda_d(\mathbf{H}) \leq \lambda_1(\mathbf{H}) \leq 3$. Given a binary OMv query vector $\mathbf{z}^{(t)}$, we construct $\overline{\mathbf{z}}^{(t)} = (\mathbf{0}_d, \mathbf{z}^{(t)}) \in \mathbb{R}^{2d}$. Since $\mathbf{H}$ has a constant condition number, we can prove that rounding an $\epsilon \sim 1/d^2$-approximate answer $\widehat{\mathbf{y}} \approx_\epsilon \mathbf{H} \cdot \overline{\mathbf{z}}^{(t)}$ still gives the correct binary answer to $\mathbf{B}\mathbf{z}^{(t)}$. □

**Reducing OMv to dynamic least-squares regression**    We next wrap up the proof of Theorem 4.2 by reducing OMv to dynamic $\epsilon$-LSR.

*Proof Sketch of Theorem 4.2.* Given a PSD matrix $H$ and a sequence of query $\{\mathbf{H}\mathbf{z}^{(t)}\}_{t=1}^T$ of the problem in Lemma 4.3, we reduce it to a dynamic $\epsilon$-LSR, where the initial $\mathbf{A}$ is such that $\mathbf{A}^\top \mathbf{A} = \mathbf{H}^{-1}$ (this preprocessing step of the reduction takes $\sim d^\omega$ time), and the label is 0 for the initial $d$ data. For each $t \in [T]$, the incoming row $\mathbf{a}^{(t)}$ is a small scaled version of $\mathbf{z}^{(t)}$, i.e, $\mathbf{a}^{(t)} = \frac{1}{d^2\sqrt{T}} \cdot \mathbf{z}^{(t)} \in \mathbb{R}^d$, and the label is 1. Let $\mathbf{x}_\star^{(t)}$ be the optimal solution at the $t$-th step, $\mathbf{H}^{(t)} := ((\mathbf{A}^{(t)})^\top \mathbf{A}^{(t)})^{-2}$, the reduction is complete via the following three steps:

- Step 1. $\mathbf{x}^{(t)}$ and $\mathbf{x}_\star^{(t)}$ are close, i.e., $\mathbf{x}^{(t)} = \mathbf{x}_\star^{(t)} \pm O(\frac{1}{d^4\sqrt{T}})$.

- Step 2. $\mathbf{x}^{(t)} - \mathbf{x}^{(t-1)}$ recovers $\mathbf{H}^{(t-1)}\mathbf{a}^{(t)}$, i.e., $\mathbf{x}^{(t)} - \mathbf{x}^{(t-1)} = \mathbf{H}^{(t-1)}\mathbf{a}^{(t)} \pm O(\frac{1}{d^4\sqrt{T}})$.

- Step 3. $\mathbf{H}^{(t-1)}\mathbf{a}^{(t)}$ is close to $\mathbf{H}\mathbf{a}^{(t)}$, i.e., $\mathbf{H}^{(t-1)}\mathbf{a}^{(t)} = \mathbf{H}\mathbf{a}^{(t)} \pm O(\frac{1}{d^6\sqrt{T}})$.

In particular, let $\mathbf{y}^{(t)} = d^2\sqrt{T}(\mathbf{x}^{(t)} - \mathbf{x}^{(t-1)})$, Step 2 and 3 directly implies $\|\mathbf{y} - \mathbf{H}\mathbf{z}^{(t)}\|_2 \leq O(1/d^2)$. This completes the proof. □

## 5    EXPERIMENTS

Our method is most suitable for data distributions that are non-uniform. Indeed, if the data has low coherence (they are all similar to each other), then the naive uniform sampling is as good as leverage score sampling. We perform empirical evaluations on our algorithm over both synthetic and real-world datasets.

**Synthetic dataset**    We follow the empirical study of (Dobriban & Liu, 2019) and generate data from the *elliptical model*. In this model $\mathbf{a}^{(t)} = w^{(t)}\Sigma\mathbf{z}^{(t)}$, where $\mathbf{z}^{(t)} \sim N(0, \mathbf{I}_d)$ is a random Gaussian vector, $\Sigma \in \mathbb{R}^{d \times d}$ is a PSD matrix, and $w^{(t)}$ is a scaler. The label is generated as $b^{(t)} =$

$\langle \mathbf{a}^{(t)}, \mathbf{x}^\star \rangle + w^{(t)}\xi^{(t)}$, where $\mathbf{x}^\star \in \mathbb{R}^d$ is a hidden vector and $\xi \sim N(0,1)$ is standard Gaussian noise. This model has a long history in multivariate statistics, see e.g. (Martin & Maes, 1979). In our experiments, we set $\Sigma = \mathbf{I}_d$ for simplicity. In order to make the dataset non-trivial, we set $w^{(t)}$ to be large $(= \sqrt{T})$ for a few $(= d/10)$ iterations, and small $(= 1)$ for the rest of the iterations. We set $T = 400000$ and $d = 500$.

**Real-world dataset**  We use the VirusShare dataset from the UCI Machine Learning Repository[4]. We select this dataset because it has a large number of features and data points, and has low errors when fitted by a linear model. The dataset is collected from Nov 2010 to Jul 2014 by VirusShare (an online platform for malware detection). It has $T = 107888$ data points and $d = 482$ features.

**Baseline algorithms**  We compare with three baseline methods. 1. *Kalman's approach* makes use of Woodburry identiy and gives an exact solution. 2. The *uniform sampling* approach samples new rows uniformly at random. 3. The *row sampling* approach samples new rows according to the exact online leverage scores. (Cohen et al., 2020)

Our experiments are executed on an Apple M1 CPU with codes written in MATLAB. We repeat all experiments for at least 5 times and take the mean. On both datasets, we initiate the model based on the first 10% of the data. The experiment results are formally presented in Figure 1 and more details can be found in Appendix E. Our algorithm consistently outperforms baseline methods: Our algorithm runs faster when achieving comparable error rates.

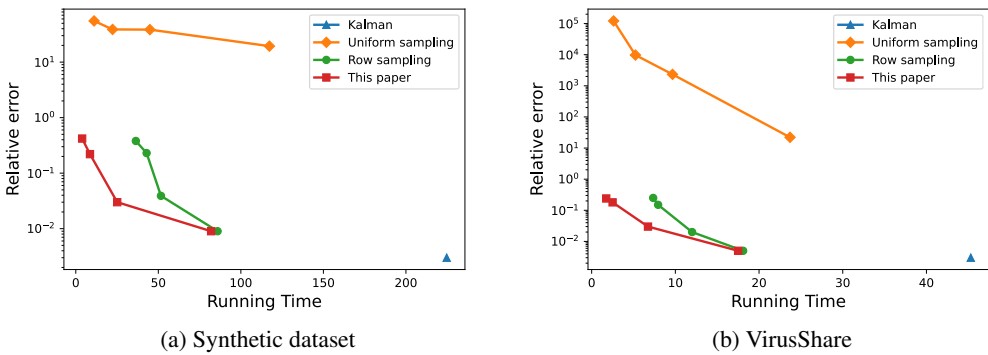

|           |
|-----------|
| (a) Synthetic dataset    (b) VirusShare |

Figure 1: Experiment results. The $x$-axis shows the running time (unit: seconds), and the $y$-axis shows the relative error $(\mathrm{err}/\mathrm{err}_{\mathrm{std}} - 1)$, where $\mathrm{err}$ is the error of the particular approach, and $\mathrm{err}_{\mathrm{std}}$ is the error of the static Normal equation. The $y$-axis is on a log scale. For uniform sampling, we take sampling probability $p = 0.05, 0.1, 0.2, 0.5$. For row sampling and our algorithm, we take the error parameter $\epsilon = 0.1, 0.2, 0.5, 1$. Kalman's approach has a relative error of 0.

## 6  CONCLUSION

We provide the first practical and *provably* fast data structure for dynamic least-squares regression, obtaining nearly tight upper and lower bounds for this fundamental problem. On the algorithmic side, we design an $\epsilon$-approximation dynamic algorithm whose total update time almost matches the *input sparsity* of the (online) matrix. On the lower bound side, we prove that it is impossible to maintain an exact (or even high-accuracy) solution with $\ll d^{2-o(1)}$ amortized update time under the OMv conjecture. As such, this result exhibits the first separation between the static and the dynamic LSR problems.

Our paper sets forth several interesting future directions. On the theoretical side, a very interesting question is whether it is possible to reduce the additive term $d^3$ of our algorithm to matrix-multiplication time $d^\omega$? A second open problem—of interest in both theory and practice—is whether it is possible to achieve input-sparsity amortized update time in the *fully dynamic* setting, i.e., when allowing both addition and *deletion* of data rows? Finally, it would be interesting to find connections between dynamic least-squares regression and incremental training of more complicated models, such as dynamic Kernel-ridge regression and to deep neural networks.

---

[4]https://archive.ics.uci.edu/ml/datasets.php

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
