# OpenReview forum: "Dynamic Least-Squares Regression"
_ICLR.cc/2022/Conference — ICLR 2022 Submitted_

### Official Review · Reviewer_Q1Nm · 2021-10-28

**Correctness:** 4
**Technical Novelty And Significance:** 2
**Empirical Novelty And Significance:** Not applicable
**Recommendation:** 6
**Confidence:** 4

**Main Review:**

The upper bound largely imitates an earlier work by Cohen et al. The original work considers the ridge regression and thus (A^TA + lambda*I)^{-1} instead of a pure (A^T A)^{-1} in the regression. The algorithms and the proofs are all similar to Cohen et al.’s paper. But the result on the regression problem is nice.

The reduction in the proof of the lower bound is interesting and I like it, though the Online Matrix-vector conjecture feels a bit strong in its own formulation to me, which already places the regression problem in a good position for the reduction. The lower bound is a bit weak in the sense that it requires eps to be 1/poly(d), which is rather small.

The paper is well-written. The following is a few typo-level comments:
- There are a few occurrences of ‘Woodburry’ instead of ‘Woodbury’.
- Page 7, three lines above the title of Section 4: delete the “<=”. The big-O notation implies that.
- page 8, proof sketch of Theorem 4.2, line 6: (A^T A)^{-2} should be (A^T A)^{-1}?

**Summary Of The Paper:**

The paper considers solving the regression problem min_x |Ax-b|_2 in the online setting, where A in R^{n x d} and b in R^n are given row by row, one at each time. The main task is to maintain a good approximate solution x (meaning that |Ax-b|_2 <= (1+eps)*OPT) throughout this process, with the update time as small as possible. The paper shows that if there are T updates to A (the initial A may not be empty), the total update time will be O(eps^{-2} nnz(A) log T + poly(eps^{-1} d log T)).

The first term O(eps^{-1}nnz(A)log T) matches the runtime in a typical sketching algorithm up to the log T factor, while the second term poly(eps^{-1} d log T) is much better in the dependence on T than the runtime of a simple sketching algorithm, which would be poly(d/eps)*T. This means that the amortized cost of each update is much smaller when T >> d.

The paper also proves a lower bound of Omega(d^{2-o(1)}) amortized update time for T = poly(d) updates, assuming the Online Matrix-vector conjecture (which is on the amortized runtime of multiplying a matrix with a vector in the online setting).

**Summary Of The Review:**

Although there is not significant technical innovation, this paper is a solid technical paper that solves a basic linear algebraic problem in the online setting and deserves to be published.

---

> ### Author Response · Authors · 2021-11-15
> **Response to Reviewer Q1Nm.**
>
> Thanks for the insightful comments!
>
>
> > “The proof seems to require that A has full column rank and this needs to be stated explicitly”
>
> The full column rank is implied by our Assumption 2.2.
>
>
> > “The lower bound is a bit weak in the sense that it requires eps to be 1/poly(d), which is rather small.”
>
> We agree on this point, but we address that the major point of this lower bound is to show that $O(d)\cdot \mathsf{polylog}(1/\epsilon)$ is not achievable, and we don’t optimize the parameter there.

---

> > ### Comment · Reviewer_Q1Nm · 2021-11-15
> > **Thanks for the response**
> >
> > > The full column rank is implied by our Assumption 2.2.
> >
> > I'm sorry I overlooked this earlier. Thanks for the clarification!

---

### Official Review · Reviewer_X5Ds · 2021-10-28

**Correctness:** 3
**Technical Novelty And Significance:** 3
**Empirical Novelty And Significance:** 2
**Recommendation:** 6
**Confidence:** 4

**Details Of Ethics Concerns:**

No concern.

**Main Review:**

# Quality and Clarity

Overall, this paper was a joy to read. It's written clearly and is well organized. No lengthy proofs are given in the body of the paper; instead elegant and intuitive proof sketches are shown. As someone very familiar with this literature, these proof sketches were largely satisfying.

# Experiments

Experiments are present, but rather barebones, and arguably misleading at a couple points.
The experiments compare wall-clock time versus relative-error for various algorithms.
While the algorithm proposed in the paper is clearly listed, the competing algorithms are not clearly described.
> For example, when the Kalman algorithm is run, which LS subroutine is used? Does it take advantage of sparsity? How can I trust that the Kalman algorithm is run with good parameters if the code isn't even provided in the supplemental material?

Additionally, the plots in Figure 1 are misleading with respect to the runtime-vs-error guarantee of the Kalman algorithm:

> The caption notes that the Kalman algorithm has zero error, which would be far below the $10^{-3}$ error shown on the plots. As drawn, the plots suggest that Kalman takes 2x the time for 1 order-of-magnitude improvement, while the caption suggests it's more like 14 orders-of-magnitude improvement (assuming "error of 0" means "roughly machine epsilon").

The papers is focused more on the theory, so I don't consider any of this a deal-breaker. But it should be acknowledged, and the authors should find a better way to compare Kalman against the other algorithms. Some concrete ideas are listed at the bottom of the review.

# Technical Novelty and Significance

The originality and novelty of this paper is a bit nuanced. The idea of online leverage score sampling, as acknowledged by the paper, is not new. So, computational complexity is really the only part in focus. If $d$ is the dimension of the least square problem, then prior results on Dynamic Least Squares (DLS) used $O(d^2)$ amortized runtime per-update, while this paper has $O(d)$ amortized runtime per-update.

### Lemma 3.4

This performance boost is achieved by approximating leverage scores with a Johnson–Lindenstrauss sketch. Peeling back the layers of the nice presentation, this seems to be the core of the technical contribution of the paper (mainly in Lemma 3.4, a bit in Lemma 3.7). The proof sketch is compelling, though I haven't reviewed the full proof; if I have time I'll make an update on that.

### Lemma 3.7

A possible second core technical contribution is in Lemma 3.7, bounding the sum of online leverage scores. Though, in my experience, I haven't seen many leverage score bounds depend on the good conditioning of the input matrix. My impression is that leverage score sampling is generally interesting _because_ it can avoid niceness assumptions on the input matrix. If the authors could comment specifically on why the good conditioning of $M$ is necessary, and if there's clear precedent in the literature, I would appreciate it.

### Section 4

A third contribution is the conditional lower bound. While I had not heard of the OMv conjecture, it is intuitive and the construction of the lower bound from the OMv conjecture is mostly elegant. Unfortunately, there is a frustrating point in the Proof Sketch of Theorem 4.2, where the vector $x^{(t)}$ is never clearly defined, so I am not totally clear how the reduction works. The proof is still believable though.

### Overall "Sufficiency"

In my view, the technical contributions, ranked from most interesting to least interesting are:
1. The JL Sketch for Leverage Scores
2. The Lower Bound Reduction
3. The sum-of-sensitivities bound

The proofs seem correct, and all three points may be relevant to future researchers, but it just isn't obvious to me that this is _sufficient_. Since I'm familiar with this area, and many of the proofs around leverage scores, I totally could see myself using any of the three points here, so I tend toward "Marginally above accept threshold".

To be clear about my _sufficiency_ concern, let me give some examples of what I might hope to see more of:
1. I'm a little worried about the Assumption $2.2$, which ensures that the data isn't too messy. I don't think that I've seen any row-norm bounds in recently leverage score sampling papers. Not sure if I've seen these $\sigma$ bounds either (though that's plausibly hidden by the $\lambda$ parameter in $\lambda$-ridge leverage score sampling).
2. Move to ridge regression or KRR. If the JL trick also works there, that would be notably compelling, and feel like a stronger and more general theoretical guarantee. This may expand the scope a lot, so I might also lean towards bullet point three:
3. The theory is rooted in the three above-discussed claims. The paper is largely about the algorithm itself though. Flesh out the experiments more.

---
# Extra Tidbits

### Re: Experiments
Here's some specific ideas on how to make Figure 1 a bit more honest:
- Maybe remove the log-scale?
- Maybe use an iterative LS algorithm that can trade-off runtime and accuracy, to compare against the proposed algorithm for a wider variety of time points?
- Maybe just draw a vertical line at 224, where the Kalman algorithm achieves zero error?
- Maybe break the y-axis and show that Kalman is much much lower, roughly around machine epsilon?

### Lingering technical questions
1. [Page 3, just before Section 1.2] Why is this $d^{1 + o(1)}$? I get that $nnz(A^{(T)}) = O(dT)$, but I don't see where the $o(1)$ term comes from.
1. [Page 7, second-to-last sentence of the first paragraph]: I believe line 4 of `UpdateMembers` takes $O(s^2 d)$ time to compute $Hm$. Shouldn't this be added into the big-O notation here? How does this interact with the amortized runtime?

### Typos and suggested edits
I found a couple typos, but very few.
1. [Top of page 3]: Right after "$T \gg d$" add a citation
1. [Page 3, third line of "Sketching and Sampling"]: "have" should be "has"
1. [Page 4, second to last line of the page]: Should be "the Woodbury Identity" not "Woodburry identity" (spelling, capitalization, and missing "the").
1. [Page 5, line 1 of Algorithm 1]: A and b should have superscripts
1. [Page 6, first bullet point in the proof sketch]: $\|B^{(t-1)}m^{(t)}\|$ should have sub- and super-scripts.
1. [Page 6, last bullet point of the proof sketch]: Consider giving some intuition why approximately preserving leverage scores implies that the spectral embedding holds, or cite a prior result which gives such an intuition? In particular, why is this a deterministic guaranteed conditioned on equation (4) holding?
1. [Page 7, and maybe some other pages too]: What is $\delta$? I assume it's a JL-parameter, but I think it's never defined.

**Summary Of The Paper:**

The paper discusses _Dynamic Least Squares_: the problem where the rows of a overdetermined least squares problem are revealed incrementally, and an algorithm has to maintain an accurate solution to the least squares system as these rows are revealed.

In prior work, this problem has been studied in the context of space complexity, where tools like leverage score sampling minimize the number of rows an algorithm must store. In this work, the focus shifts to time complexity, where the authors focus on making those leverage score sampling algorithm more computationally efficient.

The algorithm provided is matched with a compelling conditional lower bound.

Some experiments are provided.

**Summary Of The Review:**

The paper discusses an interesting problem, taking an algorithm optimized for space complexity, and tuning it to also benefit time complexity.

The technical core of the paper certainly exists, but I'm uncertain if it is sufficient. I lean "marginally above accept" on this point, since I could see myself referring to results these in the future, but it's just not a lot. I like the lower bound quiet a bit.

The experiments are barebones, but if the theoretical contribution is sufficient, then that's something the authors can fix up for the camera-ready version.

I'm particularly interested in discussing with other reviewers if the technical contributions are sufficient. I could easily be swayed either way.

---

> ### Author Response · Authors · 2021-11-15
> **Response to Reviewer X5Ds**
>
> We thank the reviewer for the insightful and detailed comments. Our response is presented below for reference. For comments not addressed below, we agree with the suggestion and will implement the change in the next draft.
>
>
> > “For example, when the Kalman algorithm is run, which LS subroutine is used? Does it take advantage of sparsity? How can I trust that the Kalman algorithm is run with good parameters if the code isn't even provided in the supplemental material?”
>
> A detailed description of Kalman’s algorithm is presented in Appendix B.2. It is a deterministic algorithm with no parameters to tune. We didn’t take advantage of sparsity in all of the experiments, so it should be fair. Full details of experiments and parameter choice can be found in the Appendix E. We can update a new version if necessary and plan to release the code.
>
>
>
> > “The caption notes that the Kalman algorithm has zero error, which would be far below the 10−3 error shown on the plots. As drawn, the plots suggest that Kalman takes 2x the time for 1 order-of-magnitude improvement, while the caption suggests it's more like 14orders-of-magnitude improvement (assuming "error of 0" means "roughly machine epsilon")”
>
> Thanks for the suggestions. We want to mention that the absolute loss of Least-squares regression does not reach zero on both experiments. In our plots we show the relative error defined by (err / err_{std} - 1), and the Kalman algorithm has 0 relative error since its error is the same as the static Normal equation. A relative error of $10^{-3}$ is already close to perfect, e.g. it makes not too much difference between 1% error and 1.001%.
>
>
> > “If the authors could comment specifically on why the good conditioning of M is necessary, and if there's clear precedent in the literature, I would appreciate it.”
>
> A simple example illustrating why good conditioning is necessary is that if the $\ell_2$ norm of the new row is doubled every time, then it is necessary to include the new row and run Kalman’s update. On the other side, we note that our dependency (and the optimal dependence) on the condition number only scales logarithmically.
>
>
>
>
>
> > “Unfortunately, there is a frustrating point in the Proof Sketch of Theorem 4.2, where the vector x(t)is never clearly defined, so I am not totally clear how the reduction works”
>
> $x^{(t)}$ is the solution maintained by the dynamic algorithm.
>
>
> > “I'm a little worried about the Assumption 2.2, which ensures that the data isn't too messy. I don't think that I've seen any row-norm bounds in recently leverage score sampling papers. Not sure if I've seen these σ bounds either (though that's plausibly hidden by the λ parameter in λ-ridge leverage score sampling).
>
> Indeed, the $\sigma$ bound naturally corresponds to the $\lambda$ parameter in $\lambda$-ridge regression. The dynamic least-squares regression problem becomes equivalent to a dynamic version of the $\lambda$-ridge regression problem when the first $d$ rows of $A$ is $\lambda * I$.
> As we mentioned before, our algorithm only has logarithmic dependencies on $\sigma$ and the norm bound $D$ (which is inevitable in some sense), so there is no large overhead.
>
> > “Move to ridge regression or KRR. If the JL trick also works there, that would be notably compelling, and feel like a stronger and more general theoretical guarantee. This may expand the scope a lot, so I might also lean towards bullet point three:”
>
> Our technique indeed works for ridge regression, by letting the first $d$ rows of $A$ to be $\lambda * I$. Assuming “KRR” =kernel ridge regression, then our method does not seem to directly generalize to kernelization (in fact, it is not even clear how to extend Karman’s classic approach in this setting). Nevertheless, this is an interesting route forward from our work.
>
>
>
> > “Maybe use an iterative LS algorithm that can trade-off runtime and accuracy, to compare against the proposed algorithm for a wider variety of time points?”
>
> We are not aware of other iterative LS methods that can trade-off runtime and accuracy. The only other options we are aware of are uniform sampling and the online row sampling approach (which are both included in our experiments).
>
> > “[Page 3, just before Section 1.2] Why is this d^{1+o(1)}? I get that nnz(A(T))=O(dT), but I don't see where the o(1)term comes from.”
>
> Sorry for the confusion, we aim to use $d^{o(1)}$ to hide logarithmic factors, and we will change it to $\tilde{O}(d)$.
>
> > [Page 7, second-to-last sentence of the first paragraph]: I believe line 4 of Updatemembers takes O(s^2d) time to compute Hm. Shouldn't this be added into the big-O notation here? How does this interact with the amortized runtime?
>
> This step takes $O(d^2)$ since it computes a matrix-vector product with matrix size $d\times d$. For amortized runtime, since UpdateMemers are only called for $O_{\epsilon}(d)$ times, the total time is at most $O_{\epsilon}(d^3)$.

---

> > ### Comment · Reviewer_X5Ds · 2021-11-29
> > **Thanks for the response**
> >
> > Thanks to the authors for the response. I like the technical updates, and many of my little confusions have been cleared up. Some summary:
> > - My concern about the representation of the Kalman algorithm in the experiments remains valid in my perspective. Full argument is below. I expect the authors to fix this editorial notion in their plots.
> > - My concerns about dependence on condition are assuaged. The dependence is logarithmic, and intuitively necessary as the authors describe in their response (I recommend mentioning this justification in the text though).
> > - My concerns about sufficiency remain. I agree with Reviewer aakT and Q1Nm that the OMv lower bound is compelling and not commonly used in this subfield. I just struggle with putting this paper clearly above the sufficiency line. It is for this reason that I maintain a score of 6. If 7 was allowed at ICLR, I would use a score of 7.
> >
> > ---
> >
> > >  A relative error of  is already close to perfect, e.g. it makes not too much difference between 1% error and 1.001%.
> >
> > I see what you mean, but I'd still say the plot is misleading. I would at least substitute the point with a vertical line. There's still a narrative being suggested when you lock $10^{-3}$ as "good enough" instead of $10^{-16}$. If you defined $10^{-6}$ as "good enough", then this would look a lot less good for you. At $10^{-16}$ it looks really bad for you. The current narrative says that _if you really need very small relative error then sampling truly is good enough_, while this might not be fair in some settings. You should reduce the amount of editorial in your plots. Anyone reading the plots less closely than me would be expected to miss this subtlety in defining an arbitrary $10^{-3}$ as good enough.
> >
> > > method does not seem to directly generalize to kernelization
> >
> > If not kernalization, perhaps another extension could be ensuring that Lewis Weights are preserved in a dynamic setting, so you could compute $\ell_p$ regression at each time? This could likely be built as a layer on top of leverage scores. It's not clear what the baseline would be, which makes this even a bit more compelling. Just another idea for a further extension of this work.
> >
> > > We are not aware of other iterative LS methods that can trade-off runtime and accuracy
> >
> > Don't all iterative LS methods trade-off runtime and accuracy? Less iterations implies both less runtime and less accuracy. Conjugate gradient is a concrete example.

---

### Official Review · Reviewer_aakT · 2021-10-30

**Correctness:** 4
**Technical Novelty And Significance:** 3
**Empirical Novelty And Significance:** 1
**Recommendation:** 8
**Confidence:** 4

**Main Review:**

It seems the exposition is largely focused on the upper bound, which achieves input sparsity runtime by performing online row sampling and and then running a rank-one update procedure to the maintained solution each time a row is sampled. However, because the number of sampled rows is roughly $\tilde{O}(d)$, I believe to get input sparsity runtime, it suffices to run the simpler algorithm of just running online row sampling and resolving the regression problem from scratch each time a row is sampled. The online row sampling algorithm uses input sparsity runtime and solving the regression problem from scratch on a matrix with size $\tilde{O}$ rows up to $\tilde{O}(d)$ times will use a lower order runtime. Thus it seems the main contribution of this paper is to improve the lower order runtime through the rank-one updates. Please let me know if I'm missing something.

On the other hand, I find the lower bound more compelling as it uses tools from fine-grained complexity that may not be as familiar to the sublinear algorithms community. The paper shows a reduction from the online matrix-vector conjecture, which essentially says that maintaining the product of a matrix and a vector whose entries are iteratively revealed requires roughly $\Omega(d^2)$ update time. This lower bound thus shows linear regression problem is strictly harder in the dynamic setting than the static setting.

The paper also includes some experiments but the details are not entirely clear. For example, it was stated that the experiments were all repeated at least 5 times. Does Figure 1 then plot the mean relative error or the best relative error? Can you elaborate more on how you chose to implement online row sampling, e.g., how you chose to approximate the online leverage scores?

Minor: Woodburry -> Woodbury in a few places

Pg. 8: scaler -> scalar

**Summary Of The Paper:**

This paper consider linear regression in the dynamic setting with the goal of minimizing the amortized time of outputting a $(1+\epsilon)$-approximate solution at all times and gives two main results. The first result is an input sparsity runtime algorithm that uses online row sampling while second result is a lower bound that shows that on average, $\Omega(d^2)$ time is needed per operation.

**Summary Of The Review:**

In summary, I think the main result in this extended abstract is a lower-order term improvement over the naive algorithm and I found the analysis unnecessarily complicated in order to achieve this lower-order improvement over the naive algorithm (please correct me if I am wrong!). On the other hand, the lower bound shows a nice separation from the centralized setting and although not entirely surprising, the result is interesting to me from a theoretical perspective.

Post-rebuttal update: I acknowledge that I have read the author response to my review. I also acknowledge that I did indeed misunderstand the significance of the upper bound with respect to the previous work. Specifically, because the JL-trick can only be applied to batches of $d$ rows, then it is not applicable in the dynamic setting in which the solution must be updated after each row. As I already thought the lower bound was of independent interest, I raise my score from 5 (weak reject) to 8 (accept).

Regardless, I hope the authors will consider improving the presentation of the technical statements. In particular, Lemma 3.3 could either decrease the number of closed-form formulas or add intuition to some of the formulas.

---

> ### Author Response · Authors · 2021-11-15
> **Response to Reviewer aakT**
>
> We thank the reviewer for the detailed comments.
>
> > “ I think the main result in this extended abstract is a lower-order term improvement over the naive algorithm and I found the analysis unnecessarily complicated in order to achieve this lower-order improvement over the naive algorithm (please correct me if I am wrong!)”
>
> We would like to correct a substantial mis-understanding of our main result here: As stated in the paper, our algorithm has a total runtime of $O_{\epsilon}(d^3 + \mathsf{nnz}(A)) \leq O_{\epsilon}(d^3+ nd)$. Therefore, the amortized runtime per iteration is $O(d)$ in the most common regime $n \gg d$. On the other hand, the total runtime of Kalman’s approach as well as Cohen’s approach is $O(nd^2)$, resulting in an amortized runtime of $O(d^2)$ per iteration. Hence, our work obtained a quadratic improvement on the amortized running time ($O(d^2)$ --> $O(d)$).
> We do not consider this as a “low-order improvement”, and we believe there might have been a misunderstanding of our result here.
>
> For clarity, let us elaborate on why Cohen el al.’s algorithm indeed has $O(d^2)$ cost per iteration: One of the main bottlenecks of their algorithm actually comes from the computation of online leverage scores, which boils down to computing $(m^{(t)})^{\top}((M^{(t-1)})^{\top}M^{(t-1)})^{-1}m^{(t)}$ in *every* iteration, thus incurring an $O(d^2)$ cost per iteration. (Please note that the paper already contains this comment, at the bottom of page 4, but we appreciate the reviewer’s comment and will make this clearer).
>
> We kindly pledge the reviewer to reconsider the merit of the paper in light of this (potential) misunderstanding.
>
>
>
>
> > “The paper also includes some experiments but the details are not entirely clear. For example, it was stated that the experiments were all repeated at least 5 times. Does Figure 1 then plot the mean relative error or the best relative error? Can you elaborate more on how you chose to implement online row sampling, e.g., how you chose to approximate the online leverage scores?”
>
> We repeat the experiments for at least 5 times and take the mean relative error.  When implementing the online row sampling, we take the same parameter for both online row sampling and our algorithm. That is, we use the same scaling factor for Line 2 of Algorithm 3. When implementing our algorithm, we additionally need to approximate the online leverage score and choose a suitable size of JL matrix. We keep the size of the JL matrix fixed through all the experiments (we slightly tuned this parameter between three choices). Full details of experiments and parameter choice can be found in the Appendix E. We can update a new version if necessary and plan to release the code.

---

> > ### Comment · Reviewer_aakT · 2021-11-15
> > **Acknowledgement of Response**
> >
> > Thanks for the clarification. I have updated my review accordingly.

---

### Official Review · Reviewer_yBkT · 2021-11-05

**Correctness:** 4
**Technical Novelty And Significance:** 3
**Empirical Novelty And Significance:** 3
**Recommendation:** 6
**Confidence:** 3

**Main Review:**

The problem studied is very natural since data points are often not available at the same time. The solution mainly follows Cohen et al. (2020) with the difference in the leverage score computation, which is done using a matrix that results from applying the JL lemma. At a technical level this part is not extremely new. The lower bound is nice and not known to the best of my knowledge. The runtime improvement seems significant (both in theory and in practice), although the numerical experiments are limited (just 1 experiment with real data). The paper is well written in general.

I have the following minor comments.

-What is the main technical insight of this work compared to Cohen et al. (2020)? It seems that the authors of that paper considered the JL trick but in a different way using batched updates. What were they missing?

-Woodburry -> Woodbury (everywhere)

-page 2 grammar: "which assert", "this papers", "the implication Theorem"

-page 3: "Theorem 1.2 almost matches the fastest sketching-based solution": I don't think it matches it, since it has an $\epsilon^{-2}$ error dependence instead of $\log 1/\epsilon$.

-page 3 grammar: "have focused"

-page 8 "scaler"

-page 9 "identiy"

**Summary Of The Paper:**

This paper studies incremental least-squares regression, where the goal is to maintain an $(1+\epsilon)$-approximate solution to $\min_x \left\Vert Ax-b\right\Vert_2^2$ for some $A\in\mathbb{R}^{n\times d}$, under row insertions to $\begin{pmatrix}A & b\end{pmatrix}$, while keeping the total runtime as low as possible.

It builds on a work of Cohen et al. (2020), where it is shown that if, upon insertion of a row, we only keep (a multiple of) it with probability roughly equal to its current leverage score (otherwise we discard it), then the total number of rows that will be inserted is only $\widetilde{O}(d)$ and the matrix spectrally approximates the "true" matrix where all rows are kept. The main contribution of this paper is to make this algorithm faster, by improving the leverage score computation procedure. Instead of directly computing the leverage scores, the authors use the Johnson-Lindenstrauss lemma to be able to quickly approximate them. Then the main thing to be taken care of is maintaining certain dimension-reduced matrices that arise. In total the asymptotic amortized runtime goes down from $\widetilde{O}(d\cdot nnz(A))$ to $\widetilde{O}(nnz(A))$.

Additionally, the authors give a negative result, which shows that, under the OMv conjecture, the amortized runtime is $\Omega(d^2)$ if we require a high-precision solution ($\epsilon=1/\mathrm{poly}(d)$).

The theoretical result is accompanied by some synthetic and real experiments. In both, it seems that the runtime improvement is significant (e.g. up to 4x speedup in the real data when $\epsilon=1$).

**Summary Of The Review:**

A significant improvement of the theoretical runtime bound is given. Some of the technical content overlaps somewhat with previous work. The experiments show good results but are limited. The paper is well written. Therefore, I tend to acceptance.

-----------------------------------------

I thank the authors for the answers, they have answered my questions.

---

> ### Author Response · Authors · 2021-11-15
> **Response to Reviewer yBkT**
>
> We thank the reviewer for the insightful comments.
>
> > “What is the main technical insight of this work compared to Cohen et al. (2020)? It seems that the authors of that paper considered the JL trick but in a different way using batched updates. What were they missing?”
>
> Cohen et al. considered *batch* processing of rows: They batch O(d) rows, and compute the approximate leverage scores of these rows all at once. Using the JL trick they can compute all these approximate leverage scores using $O(\log(1/\delta))$ calls to linear system solvers (~= $O(d^{\omega} * \log(1/\delta))$ time).
>
> For our problem, we cannot afford to use batching -- This is because whenever we receive a new row, we need to immediately output an approximate solution to the regression problem, before seeing the next row.
>
> Our main technical insight is that by maintaining a set of carefully designed matrices (certain inverse matrices and their JL-“compressed” versions), allows to quickly implement both leverage score approximations and rank-1 updates (= row insertion) to the matrix, which in turn implies an efficient dynamic update rule for the solution matrix.
>
> >  “page 3: Theorem 1.2 almost matches the fastest sketching-based solution": I don't think it matches it, since it has an eps^{-2} error dependence instead of log⁡1/ϵ.”
>
> Right -- Here $\epsilon$ is regarded as a constant (i.e., the “low accuracy” regime), hence we mean the dependency on n,d matches the static solution. We shall make this point clearer.
>
> We thank the reviewers for the grammatical corrections, and will modify them in the next version.

---

### Decision · Program_Chairs · 2022-01-20

**Decision:**

Reject

**Comment:**

There wasn't enough enthusiasm to push this paper over the bar, based on no reviewer championing the paper (the one score above 6 was consulted and thought this was a fair assessment). The reviewers appreciated the contributions of the paper but felt that in terms of technical depth, there was a lot of overlap with prior work, and the statements of the results themselves were good but not exciting enough to convince the reviewers. Some suggestions for further improvement that came up were to try to extend this to update time for low rank approximation, which was an application that other work that built off of Cohen et al did, see, e.g., https://arxiv.org/abs/1805.03765 . Regarding presentation, it would be great if in a re-submission the authors handle the presentation concerns of some of the reviewers regarding the experiments.